# In Vitro Screening of Non-Antibiotic Components to Mitigate Intestinal Lesions Caused by *Brachyspira hyodysenteriae*, *Lawsonia intracellularis* and *Salmonella enterica* Serovar Typhimurium

**DOI:** 10.3390/ani12182356

**Published:** 2022-09-09

**Authors:** Nienke de Groot, Mariana Meneguzzi, Barbara de Souza, Matheus de O. Costa

**Affiliations:** 1Departamento de Producción Animal, Facultad de Veterinaria, Universidad de Murcia, 30100 Murcia, Spain; 2Department of Veterinary Population Medicine, College of Veterinary Medicine, University of Minnesota, St. Paul, MN 55455, USA; 3Departamento de Clínica and Cirurgia Veterinárias, Escola de Veterinária, Federal University of Minas Gerais, Belo Horizonte 31270-901, Brazil; 4Large Animal Clinical Sciences, Western College of Veterinary Medicine, University of Saskatchewan, Saskatoon, SK S7N 5B4, Canada; 5Department of Population Health Sciences, Faculty of Veterinary Medicine, Utrecht University, 3584 CS Utrecht, The Netherlands

**Keywords:** swine, in vitro organ culture (IVOC), intestinal health, pathogen, feed additive

## Abstract

**Simple Summary:**

The prevention, treatment, and control of swine dysentery, ileitis, and porcine salmonellosis diseases, respectively, caused by infection with *Brachyspira hyodysenteriae*, *Lawsonia intracellularis*, and *Salmonella enterica* serovar Typhimurium, still relies on the use of antimicrobials. The goal of this study was to evaluate the effectiveness of four commercially available non-antimicrobial compounds in preventing lesions caused by these bacteria using an in vitro intestinal culture model. The findings suggest that the non-antimicrobial compounds studied may have beneficial effects for the host based on the explant model data shown. These findings represent a step towards finding alternatives to antimicrobials usage and control of swine diseases in pork production.

**Abstract:**

Swine dysentery, ileitis, and porcine salmonellosis are production-limiting diseases of global importance for swine production. They are caused by infection with *Brachyspira hyodysenteriae*, *Lawsonia intracellularis*, and *Salmonella enterica* serovar Typhimurium, respectively. Currently, the prevention, treatment, and control of these diseases still relies on antimicrobials. The goal of this study was to evaluate the effectiveness of four commercially available non-antimicrobial compounds in preventing lesions caused by the bacteria cited above using an in vitro intestinal culture model. A total of five pigs per pathogen were used and multiple compounds were evaluated. For compound F (a fungal fermented rye), S (a blend of short and medium chain fatty acids), and P (a synergistic blend of short and medium chain fatty acids, including coated butyrates), a total of four explants/pig for each treatment were used, while for compound D (an extract of carob and thyme) only 12 explants/pig for each treatment were used. Explants were exposed to a combination of pathogen only (n = 4/compound/pig), compound only (n = 4/compound/pig), or pathogen and compound (n = 4/compound/pig) and sampled at two time-points. Histopathology and gene expression levels were evaluated to investigate the treatment effect on explants. Short and medium-chain fatty acids, and an extract of carob and thyme, was found to mitigate lesions due to *B. hyodysenteriae* exposure. A fungal fermented prebiotic increased healthy epithelial coverage when explants were exposed to *L. intracellularis* or *S.* Typhimurium. These findings represent a step towards finding alternatives to antimicrobials usage and control of swine dysentery, ileitis, and salmonellosis in pork production.

## 1. Introduction

Swine dysentery (SD), ileitis, and porcine salmonellosis are intestinal diseases of grower and finisher pigs that lead to major economic losses due to poor growth performance, and increased production costs associated with treatment [1,2,3]. SD, characterized by mucohaemorrhagic diarrhea and colitis, is caused by *Brachyspira hyodysenteriae*. Recently, *B. hampsonii* and *B. suanatina* were found to be associated with a syndrome indistinguishable from SD [4,5]. Diarrhea caused by *Lawsonia intracellularis* is characterized by two clinical presentations: porcine intestinal adenomatosis (PIA) is the classic proliferative enteropathy and characterized by mucosal thickening at the chronic stage of disease, mainly affecting post-weaned pigs (between 6 and 20 weeks of age). Porcine hemorrhagic enteropathy (PHE) is the acute manifestation characterized by severe intestinal hemorrhage and melena during the acute stage, most commonly observed in young adult pigs (from 4 to 12 months of age) [1,6]. *Salmonella enterica* serovar Typhimurium leads to enterocolitis and watery diarrhea (mainly in grower and finisher pigs) [2]. Overtime several different vaccine development approaches have been explored for SD, such as bacterins [7,8,9], protein digests of whole cell bacterins [10,11], and reverse vaccinology [12]. However, these attempts failed to induce a robust immune response, and currently there is no efficient vaccine against SD commercially available [9]. In contrast, there are commercial vaccines for ileitis and salmonellosis [13,14,15]. Live and inactivated *L. intracellularis* vaccines have their own practical barriers for implementation [16,17]. *Salmonella* spp. vaccination programs are still a challenge due to the great diversity of serovars in commercial pigs, the lack of cross-protection between serovars, and the fact that vaccination can interfere with serological monitoring programs [14,18,19]. Therefore, treatment and control of these diseases under production settings still requires antimicrobial use.

The injudicious use of antimicrobials selects for resistant bacterial strains, imposing a risk for human and animal health [16,20,21,22]. Restrictions imposed on antimicrobial drugs available for veterinary use demands improved on-farm management measures, biosecurity practices, and the development of novel non-antimicrobial alternatives to treat and prevent infectious diseases [23,24,25,26]. Organic acids (OA), being short chain fatty acids (SCFA) and medium chain fatty acids (MCFA), prebiotics, phytobiotics, and enzyme inhibitors are currently being explored commercially as alternatives to antimicrobials [27,28,29,30,31,32,33,34].

The objective of this study was to evaluate the effect of five non-antimicrobial compounds (D, phytobiotic; F, prebiotic; P, blend of SCFA and MCFA; and S, blend of SCFA and MCFA) to prevent lesions following ex vivo infection of swine colon with *B. hyodysenteriae*, *L. intracellularis*, or *S.* Typhimurium.

## 2. Materials and Methods

### 2.1. Spiral Colon Collection and Explant Culture

A total of 20 healthy, commercial crossbred male pigs from high health herds 6 weeks of age were used as tissue donors. Out of 20 animals, 5 were used for *B. hyodysenteriae*, 5 pigs for *L. intracellularis*, and 5 pigs for *S.* Typhimurium. Additionally, five pigs were used for *B. hyodysenteriae* to screen for compound D only. Following euthanasia (captive bolt followed by exsanguination), distal spiral colon collection, and culture followed the protocol previously described [35]. For each pig, after gastrointestinal post-mortem examination, a lesion-free 10 cm segment of the spiral colon was aseptically collected and transported to a biosafety cabinet in a container with precooled (6–10 °C) Hank’s balanced salt solution (HBSS, VWR, Sanborn, NY, USA) within 10 min. Colon segments were washed with approximately 200 mL of the transport solution to remove luminal contents. Next, separation of the colonic serosa from the mucosa was performed on a refrigerated surface. The mucosa containing the submucosa and the muscularis mucosa was preserved and it was further divided into multiple 2 cm × 2 cm segments (explants). Each explant was individually placed with the mucosa facing up on a 70 µm cell strainer (Fisher Scientific, Hanover Park, IL, USA) in a six-well plate (Millipore Sigma, St. Louis, MO, USA) containing 3 mL of culture media (KBM-Gold calcium and phenol-red free Bullet Kit, Lonza, Walkersville, MD, USA) per well. The media volume dispensed could touch the bottom aspect of the cells strainer but not invade the inner aspect of the mucosa, therefore creating an air-liquid interface. Plates containing explants were incubated in a modular chamber (Billups Rothenberg INC, MIC101, San Diego, CA, USA) gassed for 2 min with 99% oxygen (O_2_), 1% carbon dioxide (CO_2_) gas mix. Finally, the chamber was incubated at 37 °C.

### 2.2. Inocula Preparation

The work described below was performed at the University of Saskatchewan. The study is reported in accordance with ARRIVE guidelines for in vitro studies [36]. Glass vials (9 mL) with Luria broth (LB) were used for culturing *S.* Typhimurium strain SL1344 at 37 °C. *B. hyodysenteriae* isolated from a SD case was cultured in glass vials (9 mL) with JBS broth (brain heart infusion broth supplemented with 1% (*w/v*) glucose, 5% (*v/v*) deactivated fetal bovine serum, and 5% (*v/v*) defibrinated sheep blood) anaerobically incubated using a commercial gas pack system (Oxoid AnaeroGen, Thermo Scientific, Hanover Park, IL, USA) at 39 °C with constant stirring. For *L. intracellularis*, a live vaccine strain capable of invading epithelial cells and inducing an immune response was used as inoculum (Enterisol Ileitis, Boehringer Ingelheim Vetmedica, Inc, St. Joseph, MO, USA) [37]. Immediately before inoculating explants, aliquots from each inoculum were collected for quantification kept frozen at −80 °C until processing. Inocula averaged 3.2 × 10^8^ CFU/mL for *S.* Typhimurium, 7.9 × 10^7^ genome copies/mL for *B. hyodysenteriae*, and 1 × 10^4^ cells/mL for *L. intracellularis*. *B, hyodysenteriae* and *S.* Typhimurium inocula were quantified using previously described methods (4, 76). *L. intracellularis* dose was provided by the vendor. Prior to inoculation, *B. hyodysenteriae* motility was checked as an indicator of bacteria viability using phase contrast microscopy.

For each pathogen, 1 mL of inoculum was centrifuged at 10,000× *g* for 5 min. Next, the supernatant was discarded, and the pellet was resuspended in 0.1 M, pH 7.0, sterile phosphate buffered saline (PBS). Explants in the pathogen control group (PCG) received 100 μL of inoculum of a given pathogen and explants in the combined compound control group (CCG) received 100 μL of compound only. The treatment group (TG; explant co-exposure to a given pathogen-compound combination), received 50 μL of 2× bacterial inocula and 50 μL of 2× compound dilution. After the inoculum and the compound were prepared, both were mixed and then exposure to the explants.

Compound composition and inclusion are found in Table 1. Compounds were diluted following guidelines for in vivo use. Dilutions were calculated based on explant weight to mimic the guidelines for use in vivo and confirmed to be innocuous to the mucosa by histopathology in preliminary experiments (data not shown). 

### 2.3. Challenge Trials

For *S.* Typhimurium and *L. intracellularis*, explants from five different tissue donors were evaluated and compounds F, S, and P were tested. Due to logistical reasons, 10 different pigs were used to challenge explants with *B. hyodysenteriae*: 5 were used for compound F, S, and P, and 5 additional pigs were used for compound D. For compound F, S and P; a total of 4 explants/pig for each combination group were used (Appendix A). For compound D alone a total of 12 explants/pig for each combination group were used (Appendix A). For all of the compounds, explants were randomly exposed to one of the following combination groups: (1) PCG; (2) CCG; and (3) TG. To confine the inoculum within the luminal aspect of the explants, a polystyrene ring (1 cm diameter × 1 cm height) was attached to the mucosal side of each explant using a surgical-grade cyanoacrylate adhesive (3 M Vetbond Tissue Adhesive, St. Paul, MN, USA). Due to differences in pathogen ecology, explants were co-incubated with each pathogen for the following periods: *B. hyodysenteriae* and *L. intracellularis* explants for 2 h (early time point) and 8 h (late time point), while *S.* Typhimurium explants were incubated for 45 min (early time point) and 2 h (late time point). Immediately after explant harvest at each time point, explants were fixed in 10% buffered formalin until processing for histopathology. The remaining explants per pathogen-compound combination were immersed in RNA-later (Qiagen, Germantown, MD, USA) at 4 °C for 24 h, then stored at −80 °C until PCR analysis. To confirm the absence of ante-mortem lesions, explants were preserved immediately following preparation for culture (10 min after colon collection), as described above, for histopathology and RT-PCR analyses.

### 2.4. Histopathology Analysis

Explants fixed in formalin were sectioned and stained using hematoxylin and eosin (H&E). An evaluator (MM) blinded to slide identification assessed the percentage of healthy epithelium and the mucus layer thickness (for *B. hyodysenteriae*-challenged explants only) covering explants. A digital image of each explant, covering its entire length, was analyzed using an image processing software (Image Pro, version 9.2, Media Cybernetics, Inc, Rockville, MD, USA). Healthy epithelium was defined as the superficial layer of cells covering the luminal aspect of the explants in a simple columnar fashion, without signs of metaplasia (abnormal cell shape), edema (increased intercellular space), or apoptosis and necrosis (picnotic or misshaped nuclei). One measurement of healthy epithelium covering the total length of each explant was obtained and data were reported as a percentage. Mucus layer thickness was measured at five evenly spaced locations along the length of the explant (far left, left, center, right, far right) and an average mucus layer thickness was reported for each explant.

### 2.5. Reverse Transcriptase Real-Time PCR (RT-PCR) Assays

Analyses of explant mRNA levels targeted the glyceraldehyde-3-phosphate dehydrogenase (GAPDH) as housekeeping reference gene, TNF-α genes, IFN-γ [38], and IL-1α [39]. iNOS was evaluated for *B. hyodysenteriae* [40] samples only. Total RNA load was extracted from explants preserved in RNA-later using a commercial kit (RNeasy Plus animal cell and tissue kit, Qiagen, Austin, TX, USA). Complementary DNA (cDNA) was generated following a commercial kit instruction (QuantiTect Reverse Transcription Kit, Qiagen, Germantown, MD, USA). cDNA samples were diluted with nuclease-free water to a final concentration of 500 ng/mL.

RT-PCR reactions were performed using an ABI 7500 Fast Real-Time PCR System (Applied Biosystems^®^, ThermoFisher Scientific, Hanover Park, IL, USA). Each 20 μL reaction contained 10 µL of PowerUp SYBR Master Mix^®^, 1 μL of forward and 1 μL of reverse primers (10 μM each), 6 µL of nuclease-free water and 2 µL of cDNA template. Reactions were incubated at 50 °C for 2 min and 95 °C for 2 min, followed by 40 cycles of 15 s at 95 °C, 30 s at 60 °C and 1 min at 72 °C and a melt curve step of 95 °C for 15 s, 60 °C for 1 min, and 95 °C for 15 s. Every reaction was performed in duplicate, alongside negative extractions, and no-template controls in each run. Samples where duplicates differed by more than 1 Ct were re-analyzed. GAPDH expression levels was constant across all samples.

For evaluation of the effectivity of the challenge method and the screening methods used in this study, comparisons between explants from the pathogen control group (PCG) and combined compound control groups (CCG) (compound D, F, P and S combined) for a given pathogen were performed regarding histopathology and RT-PCR assay analysis. Results are shown in the Appendix A.

### 2.6. Statistical Analysis

Mucus layer thickness and percentage of healthy epithelium data were compared between challenge groups by generalized estimating equations (GEE) using an unstructured correlated working matrix while clustering by pig. The data followed a normal distribution. Statistical analysis was performed using IBM SPSS 21 (IBM Corporation, Armonk, NY, USA). Messenger RNA levels (Ct) were analyzed using the MCMC qPCR package (one-way design) with a naive statistical model [41] on R studio (version 1.1.463) [42].

## 3. Results

A summary of Significant Findings is Presented in Table 2.

### 3.1. Brachyspira Hyodysenteriae

#### 3.1.1. Early Time Point

Explants treated with compound D (TG) trended towards higher epithelial coverage, when compared to PCG (*p* = 0.06). For compound F, explants from the PCG showed significant higher level of epithelial coverage when compared to either the CCG or treatment group (TG) (Figure 1A). Compound S and treated explants (TG) showed a trend to reduced mucus layer thickness (*p* = 0.06, Figure 1A) when compared to the PCG. Compound P significantly reduced mucus layer thickness (*p* < 0.05) compared to PCG and CCG. Treatment with compound F significantly down-regulated TNF-α mRNA expression, when compared to PCG (Figure 1C). Up-regulation of iNOS mRNA expression was observed for compound S TG when compared to explants from the PCG (Figure 1C). No difference was observed in mRNA expression for all genes investigated for compound P and D.

#### 3.1.2. Late Time Point

Surprisingly, increased epithelial coverage was found in explants from the PCG, when compared to compound F TG samples (Figure 1B). Compound P treated explants had significant higher epithelial coverage than the PCG and TG (Figure 1B). Mucus layer thickness was significantly increased in the PCG when compared to TG for compound P (Figure 1B). Explants treated with compound D had significantly lower levels of TNF-α, IL-1α, and INF-γ mRNA detected when compared to samples from the PCG. For compound P, TNF-α was found down regulated in TG samples, when compared to PCG (Figure 1C) and INF-γ mRNA level trended towards downregulation (*p* = 0.08, Figure 1C). Treatment of explants with compound S led to the up-regulation trend of IL-1α (*p* = 0.06, Figure 1C) and significant up regulation of INF-γ (Figure 1C), in relation to the PCG.

### 3.2. Lawsonia Intracellularis

#### 3.2.1. Early Time Point

Healthy epithelium coverage was significant higher in compound F CCG than PCG (Figure 2A). No differences in mRNA level were observed for any compound (Figure 2B).

#### 3.2.2. Late Time Point

Epithelium coverage for compound F was higher in the TG than PCG samples (Figure 2A). No other differences were observed for any gene-compound combination (Figure 2B).

### 3.3. Salmonella enterica Serovar Typhimurium

#### 3.3.1. Early Time Point

Explants treated with compound F had a significant higher epithelium coverage in CCG compared to PCG. Compound P CCG maintained significant higher epithelial coverage than the PCG and TG samples (Figure 3A). For compound S, IL-1α mRNA level was higher in TG than PCG (Figure 3B). Compound P treated explants had a trend to decrease levels of IL-1α mRNA (*p* = 0.07, Figure 3B), when compared to the PCG.

#### 3.3.2. Late Time Point

No significant differences were seen for compound F in the late time-point. Compound P TG and PCG had a significant lower epithelial coverage compared to the CCG samples (*p* < 0.05; Figure 3A). Explants in compound S TG had higher percentage of healthy epithelium coverage trend than the PCG (*p* = 0.07, Figure 3A). No gene expression differences were observed (Figure 3B).

## 4. Discussion

There is an increasing need for alternatives strategies to treat livestock bacterial diseases without the use of antimicrobials. In this study, we used in vitro porcine colon culture to evaluate the efficacy of non-antimicrobial compounds in preventing tissue damage following exposure to *B. hyodysenteriae*, *L. intracellularis* or *S.* Typhimurium.

Compound P treatment, a blend of MCFA and SCFA, improved explant epithelial coverage and decreased the accumulation of mucus and the expression of TNF-α mRNA following challenge with *B. hyodysenteriae* (Figure 1A). A trend towards downregulation of IFN-γ mRNA expression following challenge was also observed (Figure 1C). Compound S, a blend of OA decreased the accumulation of mucus, and reduced the expression of TNF-α and iNOS mRNA following challenge with *B. hyodysenteriae.* TNF-α and IFN-γ have a recognized role in tight junction regulation [43,44]. Tight junction proteins, such as occludins, claudins, and zonulae occludentes (ZO), are crucial for the maintenance of epithelial barrier integrity and to regulate the paracellular movement of ions and water [45,46]. Fatty acids appear to modulate tight junction permeability and have an anti-inflammatory effect in the colon [47,48,49]. Increased TNF-α and IFN-γ levels lead to the rearrangement of myosin molecules associated with tight-junction proteins, consequently increasing paracellular permeability [50,51,52]. In our study, TNF-α and IFN-γ mRNA expression was downregulated when explants were treated with a blend of MCFA and SCFA, including butyrates (compound P), while an upregulation was observed in explants being treated with compound S (which does not contain butyrates). Similar responses were identified in weaned pigs supplemented with butyrate, when culturing human colonic biopsies, in human colonic cell lines, and in isolated lamina propria cells with butyrate [53,54,55,56]. Intestinal epithelial cells exposed to TNF-α and IFN-γ have reduced cystic fibrosis transmembrane conductance regulator (CFTR) expression and chloride (Cl^−^) secretion [57,58,59]. This impairment of anion secretion affects the mucus layer integrity. Mucins require the interaction of bicarbonate (HCO^3−^) and Cl- with calcium (Ca^2+^) for proper release and expansion from goblet cells [60,61]. A recent study indicated that host cytokines are not responsible for the impairment of anion channels, and that *B. hyodysenteriae* may directly cause the decrease in Cl^−^ secretion and which may lead to mucin aggregation and accumulation [62]. In contrast, our findings suggest a relationship between reduced gene expression of TNF-α and IFN-γ and a reduction in mucus secretion following infection with *B. hyodysenteriae* and treatment with compound P. This link between host cytokines and mucus secretory response in SD remains to be clarified. In addition, it is important to highlight that the fold changes observed in this study were quantitatively small when compared to previously published data. This could be an effect of the model used and the biological significance remains to be explored.

Explants treated with compound F (prebiotic based on *Agaricus subrufescens* fermented rye) had higher epithelial coverage when challenged with *L. intracellularis* (Figure 2A) than those untreated. Riboglucans, β-glucans and glucomannans are examples of bioactive polysaccharides isolated from *A. subrufescens* [63]. These molecules can act as a substrate for bacterial adherence, as they mimic the host glycocalyx [64]. D-mannose, a prebiotic, reduced the adhesion of *Escherichia coli*, *Vibrio cholerae*, *Campylobacter jejuni*, and *S.* Typhimurium to HT-29 cells as per the concept described above [65]. This effect was also observed in animal studies, when weaned piglets feed was supplemented with *Lentinus edodes* mycelium extracts, leading to reduced viable counts of *E. coli* and Streptococci in the digesta (stomach, jejunum) and mucosal scrapings of the small intestine [66]. In vitro studies with ingredients in compound F have also proven binding affinity to *S.* Typhimurium and *S.* Enteritidis, and in vivo reducing peak and average shedding of these bacteria [67,68]. However, our data revealed no significant effect of compound F in epithelial coverage or cytokine expression following *S.* Typhimurium challenge. To the best of our knowledge, this is the first report which evaluates the effectiveness of *A. subrufescens* rye fermentation against *L. intracellularis*.

A recent study [69] showed the potential immunomodulatory effect of compound F when supplementing piglets post-weaning, with a reduction of pro-inflammatory cytokine production in jejunum, ileum, and colon. In our study, no significant differences in cytokine mRNA levels were observed after *L. intracellularis* challenge (Figure 2B). This observation may be due to the short period of in vitro incubation which may lead to a low level of bacteria infecting and propagating inside the epithelial cells. Previous authors reported that the pathogen may take up to 12 h to invade cells after oral inoculation, or 6 h when ligated intestinal loops were infected directly with vaccine inoculum [70,71]. The ability of the attenuated vaccine strain to induce such changes is also questionable, but it has been shown to do so in vivo [37]. However, the inoculum concentration used in current study for *L. intracellularis* challenge would not be considered to cause clinical disease and lesions in natural infections, and therefore can explain the lack of effect between the PCG and the CCG or for almost all TGs challenged with *L. intracellularis.* Thus, further studies investigating the immunomodulatory role of compound F following infection with a virulent *L. intracellularis* using longer incubation periods are strongly suggested.

Surprisingly, a lower degree of epithelial coverage was observed in explants exposed to compound F alone than explants exposed to *B. hyodysenteriae* (Figure 1A,B). It is known that colon explants harbor a microbiota compositionally similar to the donor pig prior to euthanasia [72]. Thus, we postulate that compound F may have served as a substrate for the microbiota already present in the explants, leading to bacterial overgrowth. The lack of colonic peristalsis may have further contributed to our observations.

Explants infected with *B. hyodysenteriae* and treated with compound D (phytobiotic) had increased epithelial coverage and decreased levels of IL-1α, TNF-α, and IFN-γ when compared to infected untreated explants (Figure 1A,C). Thymol and carvacrol are present in the essential oils extracted from thyme (*Thymus vulgaris*), the active ingredient in compound D [73]. Carvacrol was demonstrated to have a gastroprotective effect in a rodent model of gastritis [74,75]. It was associated with reduced colonic lesions in colitis induced by 2,4,6-trinitrobenzenesulfonic (TNBS) in rats [76] and in acetic acid-induced colitis in mice [77]. The protective effect of carvacrol was associated with its ability to regulate cyclooxygenase-2 (COX-2) expression [78,79]. An in vitro T cell model also linked the reduction of IL-2 and IFN-γ expression to exposure to thymol and carvacrol [80]. In contrast, IL-1β and TNF-α induce the expression of COX-2 [81]. Mice treated with carvacrol had decreased TNF-α levels and milder lesions following acetic acid-induced colitis [77]. Additionally to the effects of thyme, carob (*Ceratonia siliqua*, another ingredient in compound D) contains phenolic compounds such as flavonoids and gallotannins that also inhibit COX-2 [82]. Thus, the effect of compound D was likely due to its anti-inflammatory effects associated with the inhibition COX-2 cascade.

In our study, no significant differences in cytokine mRNA levels were observed after *L. intracellularis* challenge. This observation may be due to the short period of in vitro incubation which may have led to a low level of bacteria infecting and propagating inside of the epithelial cells. Previous authors reported that the pathogen may take up to 12 h to invade cells after oral inoculation, or 6 h when ligated intestinal loops were infected directly with vaccine inoculum [70]. The ability of the vaccine strain, at the same dose used in our study, to induce such changes is also questionable, but it has been shown to do so in vivo [37]. Thus, further studies investigating the immunomodulatory role of compound F following infection with a virulent *L. intracellularis* during longer incubation periods are strongly suggested.

## 5. Conclusions

In conclusion, our findings suggest that the non-antimicrobial compounds studied may have beneficial effects for the host based on the explant model data shown. Compound P supported epithelial survival and reduced mucus thickness when explants were exposed to *B. hyodysenteriae*. Compound D has an immune-modulating effect in explants challenged with *B. hyodysenteriae*. Compound F prevented epithelial death following *L. intracellularis* exposure. The authors believe that further investigations are warranted to verify compound effectiveness in vivo.

## Figures and Tables

**Figure 1 animals-12-02356-f001:**
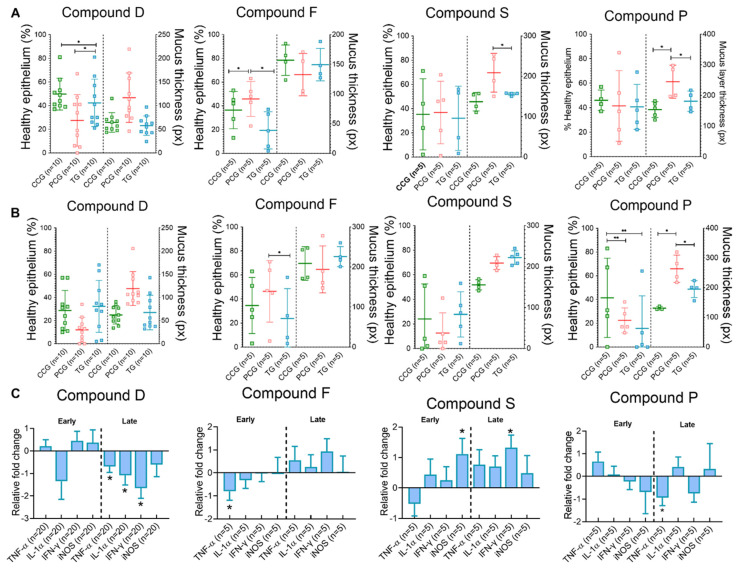
Microscopical changes and gene expression data from explants collected from 10 pigs and challenged with *B. hyodysenteriae*. (**A**,**B**) Histopathology assessment including the percentage of healthy epithelium covering explants and mucus layer thickness at early (**A**) and late (**B**) time-points. Horizontal lines represent group mean, and whiskers depict ± standard deviation from the mean. (**C**) Gene expression data is reported as fold change from TG samples using the PCG as reference. Compound control group (CCG), pathogen control group (PCG), treatment group (TG). Bars represent mean mRNA levels; whiskers depict standard deviation from the mean. Star denotes significant difference (*p* ≤ 0.05) and two stars denote *p* = 0.06.

**Figure 2 animals-12-02356-f002:**
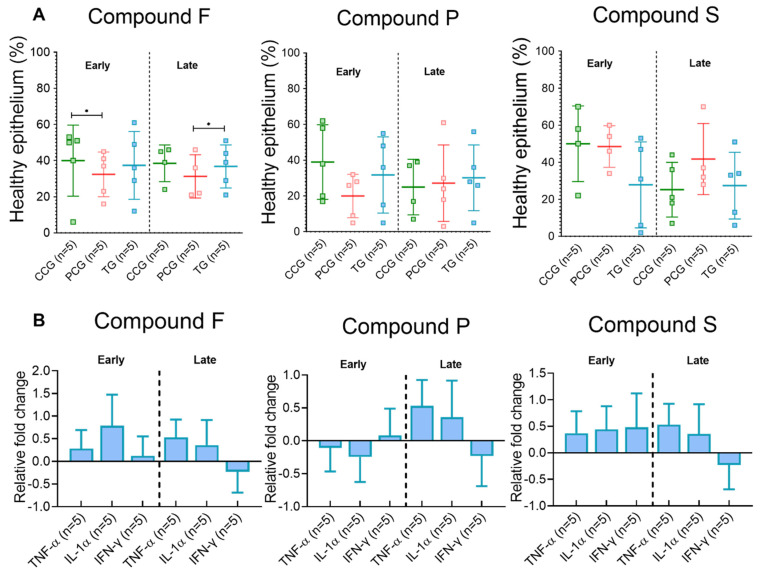
Microscopical changes and gene expression data from explants challenged with *L. intracellularis*. (**A**) Histopathology assessment. Horizontal lines represent group mean, and whiskers depict standard deviation from the mean. (**B**) Gene expression data is reported as fold change from TG samples using the PCG as reference. Compound control group (CCG), pathogen control group (PCG), treatment group (TG). Bars represent mean mRNA levels; whiskers depict standard deviation from the mean. Star denotes significant difference (*p* ≤ 0.05).

**Figure 3 animals-12-02356-f003:**
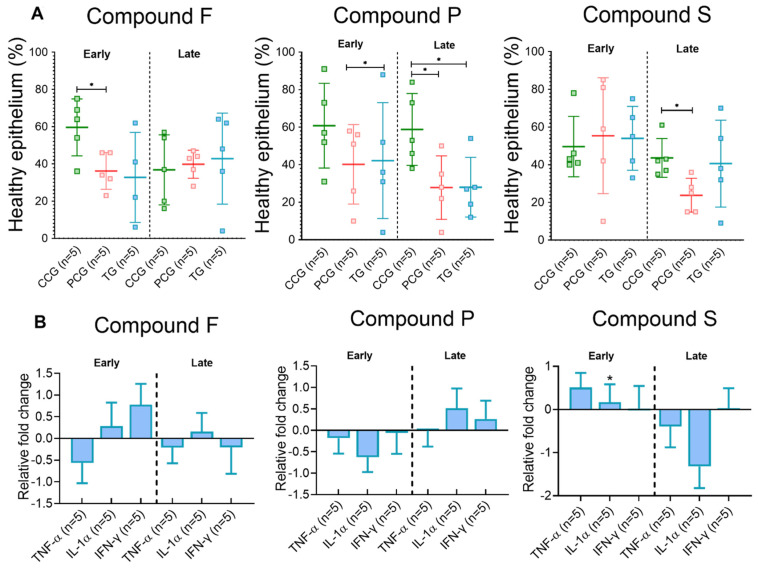
Microscopical changes and gene expression data from explants challenged with *S.* Typhimurium. (**A**) Histopathology assessment. Horizontal lines represent group mean, and whiskers depict standard deviation from the mean. (**B**) Gene expression data is reported as fold change from TG samples using the PCG as reference. Compound control group (CCG), pathogen control group (PCG), treatment group (TG). Bars represent mean mRNA levels; whiskers depict standard deviation from the mean. Star denotes significant difference (*p* ≤ 0.05).

**Table 1 animals-12-02356-t001:** Compound composition, inclusion in complete feed, and dilutions used in experiment.

Compound	Composition	Inclusion
D	Blend of thymol and carvacrol	1 kg/1000 kg of feed 0.0028 mg/g of explant
F	Fungal fermented rye	3 kg/1000 kg of feed 0.0042 mg/g of explant
P	Blend of short chain fatty acids including coated butyrates and slow-release medium chain fatty acids	3 kg/1000 kg of feed 0.0042 mg/g of explant
S	Blend of short chain fatty acids and medium chain fatty acids	3 kg/1000 kg of feed 0.0042 mg/g of explant

**Table 2 animals-12-02356-t002:** Summary of significant findings when comparing the treatment group (TG) vs. pathogen control group (PCG) across the different pathogen-compound (C) combinations.

Pathogen	Compound	Early Time-Point	Late Time-Point
*B. hyodysenteriae*	D	Increased epithelial coverage.	IL-1α, INF-γ and TNF-α down-regulated.
	F	Decreased epithelial coverage. TNF-α down-regulated.	Decreased epithelial coverage
	P	Decreased mucus layer thickness.	Decreased mucus layer thickness. TNF-α down-regulated.
	S	Decreased mucus layer thickness. iNOS up-regulated.	INF-γ up-regulated.
*L. intracellularis*	F	-	Increased epithelium coverage.
*S.* Typhimurium	P	Increased epithelialc overage.	
	S	IL-1α up-regulated.	

## Data Availability

The datasets used and/or analysed during the current study are available from the corresponding author on reasonable request.

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
