# Peer review of "In Vitro Screening of Non-Antibiotic Components to Mitigate Intestinal Lesions Caused by Brachyspira hyodysenteriae, Lawsonia intracellularis and Salmonella enterica Serovar Typhimurium"

_animals, 2022, doi:10.3390/ani12182356_

Round 1

Reviewer 1 Report

1. In the introduction you mention that more species are responsible for SD, this is also the case for Salmonellosis

2. Please give a reference for the ARRIVE guidelines, which version did you use?

3. For the LI inoculation you use the Boehringer strain, which is mitigated. I would expect a paragragh in the discussion about the effect of this mitigation.

4. Page 3 line 124 and 125: This is the first time you use PCG and CCG, please explain the abbreviation.

5. Page 3 line 147 and especially 148 are not fully clear what you mean.

6. page 5 is a long list of proza to state the results.....I do not know how to improve the presentation, but please give it a try.

7. Page 7: Sequence is different from page 6 and pages 8, 9. Please be consistent.

The conclusion that the results must be checked in vivo is obvious, but if the method described in this paper is valid and the results may be extrapolated to in vivo pigs, the impact will be sky high.

Author Response

Reviewer 1

Thanks a lot for these comments and very valuable reviews to improve its quality

  1. In the introduction you mention that more species are responsible for SD, this is also the case for Salmonellosis

In this case we only care about Typhimurium since this is the one associated with GIT signs.

  1. Please give a reference for the ARRIVE guidelines, which version did you use?

The study was designed 5 years ago (2010), therefore v1.0 was used. Indeed reference was missing in the manuscript, therefore it has been included

  1. For the LI inoculation you use the Boehringer strain, which is mitigated. I would expect a paragragh in the discussion about the effect of this mitigation.

Indeed important to refer to this mitigation. We did have included a paragraph in the discussion regarding this issue, more specifically in lines L307-309.

  1. Page 3 line 124 and 125: This is the first time you use PCG and CCG, please explain the abbreviation.

Thanks for the observation, explanations included in L126-128 in revised version of manuscript

  1. Page 3 line 147 and especially 148 are not fully clear what you mean.

To clarify these sentences I have included early and late time point for explanation of method and accordingly the timepoints per challenge

  1. page 5 is a long list of proza to state the results.....I do not know how to improve the presentation, but please give it a try.

We agree with the reviewer’s perspective. Therefore, we have included a summary of our significant findings in Table 2.

  1. Page 7: Sequence is different from page 6 and pages 8, 9. Please be consistent.

Thanks, I adapted the figure caption of Figure 1 to make them more consistent

The conclusion that the results must be checked in vivo is obvious, but if the method described in this paper is valid and the results may be extrapolated to in vivo pigs, the impact will be sky high.

We appreciate that the reviewer shared their thought and excitement with these results. We hope it will have an impact.

Reviewer 2 Report

Groot et al. evaluate the effectiveness of four commercially available non-antimicrobial compounds in preventing lesions caused by these bacteria using an in vitro intestinal culture model. Authors found that Compound P supported epithelial survival and reduces mucus thickness when explants were exposed to B. hyodysenteriae. Compound D has an immune-modulating effect in explants challenged with B. hyodysenteriae. Compound F prevented epithelial death following L. intracellularis exposure. The study is more interesting, however, more data about these products should be provided in the „Materials and methods“ section.

Other minor comments:

-        Title: In vitro, Brachyspira hyodysenteriae, Lawsonia intracellularis and Salmonella should be italic

-        Line 20: please add a period before the word „still“

-        Line 22: in vitro should be italic

-        Line 53: Please expand the full name of PIA

-        Line 55: Please expand the full name of PHE

   Line 84-86: Brachyspira hyodysenteriae, Lawsonia intracellularis and Salmonella should be italic

-        Line 103: „Compound“ should be „compound“

-        Line 246: Please delete the subtitle (3.2.) and insert the table and figures in the text where they are firstly cited.

-        Line 246: L. intracellularis  should be italic

-        Lines 379-385: Please delete this part, mentioned in the conclusions.

-        Supplementary figure 3: Please add A, B, and C on the figures?

Author Response

Reviewer 2

Groot et al. evaluate the effectiveness of four commercially available non-antimicrobial compounds in preventing lesions caused by these bacteria using an in vitro intestinal culture model. Authors found that Compound P supported epithelial survival and reduces mucus thickness when explants were exposed to B. hyodysenteriae. Compound D has an immune-modulating effect in explants challenged with B. hyodysenteriae. Compound F prevented epithelial death following L. intracellularis exposure. The study is more interesting, however, more data about these products should be provided in the „Materials and methods“ section.

Thanks for this valuable comment. The additives are commercially available however we are restricted to proprietary information regarding all the ingredients. We have therefore included a table with the composition of each compound (table 1) to the best of our knowledge.

Other minor comments:

Thanks a lot for these comments and very valuable reviews to improve its quality

-        Title: In vitro, Brachyspira hyodysenteriae, Lawsonia intracellularis and Salmonella should be italic

Adapted to italic

-        Line 20: please add a period before the word „still“

Adapted and included a period

-        Line 22: in vitro should be italic

Adapted to italic

-        Line 53: Please expand the full name of PIA

Porcine Intestinal Adenomatosis (PIA) included in line 53

-        Line 55: Please expand the full name of PHE

Porcine Haemorrhagic Enteropathy (PHE) included in line 55-56

-       Line 84-86: Brachyspira hyodysenteriae, Lawsonia intracellularis and Salmonella should be italic

Adapted to italic

-        Line 103: „Compound“ should be „compound“

Adapted to compound without a capital C

-        Line 246: Please delete the subtitle (3.2.) and insert the table and figures in the text where they are firstly cited.

Indeed the figures were not placed correctly. The figures have been moved to the point in the text where they are first mentioned

-        Line 246: L. intracellularis should be italic

Adapted to italic

-        Lines 379-385: Please delete this part, mentioned in the conclusions.

This text was indeed duplicated in the conclusions, therefore it has been removed from the discussion

-        Supplementary figure 3: Please add A, B, and C on the figures?

Thanks, I included an A B and C in Supplementary Figure 3 for clarification

Reviewer 3 Report

In general, this work is really interesting and well written. My comments aim to increase the scientific soundness and clarity of it.

Line 3 – use italics for Brachyspira hyodysenteriae and Salmonella enterica

Line 53 and 55 -  Please explain what PIA and PHE stand for ? Line 78 – ex vivo should be in italics. Line 84-85 – strains should be written in italics Line 86 – Please provide the method of euthanasia. Lines 87 – From anatomical point of view the pigs colon is divided into acending colon, transverse colon and descending colon. Please explain what is “distal spiral colon”? Line 142 – This is the first appearance of PCG, CCG and TG. Please explain its meaning here. Line 192 – it is not clear what statistical tests were used to determine differences. Results – photographic documentation of histopathological analysis is missing.

Author Response

Reviewer 3

In general, this work is really interesting and well written. My comments aim to increase the scientific soundness and clarity of it.

Thanks a lot for these comments and very valuable reviews to improve its quality

Line 3 – use italics for Brachyspira hyodysenteriae and Salmonella enterica

Adapted to italics

Line 53 and 55 - Please explain what PIA and PHE stand for ? 

Very fair point, no explanation of these abbrevations were included. Now in line 53 in included Porcine Intestinal Adenomatosis (PIA) and Porcine Haemorrhagic Enteropathy (PHE) in lines 55-56

Line 78 – ex vivo should be in italics.

Adapted to italic

Line 84-85 – strains should be written in italics 

Adapted to italic

Line 86 – Please provide the method of euthanasia. 

Good remark, I included the method of euthanasia used in the study (Captive bolt followed by exsanguination) in line 87-88

Lines 87 – From anatomical point of view the pigs colon is divided into acending colon, transverse colon and descending colon. Please explain what is “distal spiral colon”? 

In swine, the colon is commonly referred in veterinary medicine to as spiral colon, and it is divided in proximal, apex and distal spiral colon. We have used this as veterinarians are familiar with it. Also, other publications have used this terminology in the past https://pubmed.ncbi.nlm.nih.gov/3154081/ . However, if this causes confusion, we can adapt it to descending colon.

Line 142 – This is the first appearance of PCG, CCG and TG. Please explain its meaning here. 

Thanks for the observation, explanations of these abbreviations are included in L126-128 in revised version of manuscript

Line 192 – it is not clear what statistical tests were used to determine differences. 

Please refer to Lines 193-199 within the methods section for further clarification on the tests employed.

Results – photographic documentation of histopathological analysis is missing.

The infection model used here has been extensively used and characterized before through histopathology, metabolomic, metagenomic and transcriptomic analyses (Costa et al., 2016. Welle et al., 2017, Costa et al., 2020). To avoid duplicating efforts, here we have documented evidence of infection by Salmonella Typhimurium and Lawsonia intracellularis. Photographic evidence is shown in supplementary Figure 1.